# Design and Development of Low- and Medium-Viscosity Alginate Beads Loaded with Pluronic^®^ F-127 Nanomicelles

**DOI:** 10.3390/ma16134715

**Published:** 2023-06-29

**Authors:** Flora Kalogeropoulou, Dimitra Papailiou, Chrystalla Protopapa, Angeliki Siamidi, Leto-Aikaterini Tziveleka, Natassa Pippa, Marilena Vlachou

**Affiliations:** 1Section of Pharmaceutical Technology, Department of Pharmacy, School of Health Sciences, National and Kapodistrian University of Athens, Panepistimioupolis Zografou, 15771 Athens, Greece; florakalo@gmail.com (F.K.); dimpap725@gmail.com (D.P.); cprotopapa@pharm.uoa.gr (C.P.); asiamidi@pharm.uoa.gr (A.S.); 23rd Department of Internal Medicine, ‘Sotiria’ Hospital, 11527 Athens, Greece; 3Section of Pharmacognosy and Chemistry of Natural Products, Department of Pharmacy, National and Kapodistrian University of Athens, Panepistimiopolis Zografou, 15771 Athens, Greece; ltziveleka@pharm.uoa.gr

**Keywords:** hydrogels, Pluronic^®^ F-127, crosslinking, swelling studies, viscosity, alginate beads, acetyl salicylic acid, micelles

## Abstract

The anionic polymer sodium alginate, a linear copolymer of guluronic and mannuronic acids, is primarily present in brown algae. Copolymers are used in the sodium alginate preparation process to confer on the material strength and flexibility. Micelles and other polymeric nanoparticles are frequently made using the triblock copolymer Pluronic^®^ F-127. The purpose of the present study is to determine the effect of sodium alginate’s viscosity (low and medium) and the presence of Pluronic^®^ F-127 micelles on the swelling behavior of the prepared pure beads and those loaded with Pluronic^®^ F-127 micelles. The Pluronic^®^ F-127 nanomicelles have a size of 120 nm. The swelling studies were carried out at pH = 1.2 (simulated gastric fluid-SGF) for two hours and at pH = 6.8 (simulated intestinal fluid-SIF) for four more hours. The swelling of both low- and medium-viscosity alginate beads was minor at pH = 1.2, irrespective of the use of Pluronic^®^ F-127 nanomicelles. At pH = 6.8, without Pluronic^®^ F-127, the beads showed an enhanced swelling ratio for the first four hours, which was even higher in the medium-viscosity alginate beads. With the addition of Pluronic^®^ F-127, the beads were dissolved in the first and second hour, in the case of the low- and medium-alginate’s viscosity, respectively. In other words, the behavior of the mixed hydrogels was the same during the swelling experiments. Therefore, the presence of Pluronic^®^ F-127 nanomicelles and medium-viscosity sodium alginate leads to a higher swelling ratio. A model drug, acetyl salicylic acid (ASA), was also encapsulated in the mixed beads and ASA’s release studies were performed. In conclusion, the prepared systems, which are well characterized, show potential as delivery platforms for the oral delivery of active pharmaceutical ingredients and biopharmaceuticals.

## 1. Introduction

Natural polysaccharides are being investigated for potential use in the pharmaceutical and food industries due to their biocompatibility and biodegradability. Alginates, a frequent ingredient in pharmaceutical formulations and drug delivery systems, is one such polysaccharide. Alginate is an anionic polysaccharide that is naturally found in brown seaweed. Its long linear polymer chain is composed of *β*-*D*-mannuronic acid and *α*-*L*-guluronic acid, linked in a (1–4) configuration [1].

Multifunctional hydrogels based on alginates, chitosan, ulvan, and other polysaccharides have numerous applications in biomedical sciences, including drug delivery, wound dressing, tissue engineering, and temporary scaffolding for cells. Hydrogels and nanogels are particularly well suited for the controlled release of bioactive compounds, offering several advantages, such as high-physicochemical stability, responsiveness to external stimuli, high-loading efficiency, oral administration, and biocompatibility (i.e., low toxicity and immunogenicity). Stimuli-responsive hydrogels, such as glucose-responsive, pH-responsive, and thermoresponsive hydrogels, have been utilized to achieve the controlled and sustained release of active pharmaceutical ingredients, proteins, and peptides [2,3]. The swelling behavior stands out as the primary parameter requiring investigation to develop alginate hydrogels as delivery vehicles for water-soluble drugs and proteins for oral consumption. 

pH-sensitive hydrogels can be prepared using polymers obtained from both natural and synthetic sources. Sodium alginate has a high capacity to chelate divalent cations such as Ca^2+^, leading to the formation of a 3D network with a distinctive “egg-box” structure [2,4]. The alginate beads exhibit remarkable pH sensitivity owing to the abundant carboxyl groups that become protonated under acidic conditions but ionized in alkaline environments. Alginate beads have been a widely investigated and produced material for drug delivery systems intended for oral administration because of their pH sensitivity and outstanding ability to create gels [4]. In one study, alginate, calcium chloride, and chitosan were ionotropically gelled to create bovine serum albumin-loaded beads. Particle size and encapsulation effectiveness of chitosan–alginate beads were strongly influenced by the chitosan content. At the higher pH, the chitosan–alginate matrices quickly dissolved, causing a burst release of the protein [5]. According to the recent literature, an acetic acid coagulation bath and Ca^2+^ chelation were used to make nanocellulose and sodium alginate miscible, creating an interpenetrating network hydrogel via double crosslinking with considerable prolongation of the aspirin release period to around 60 h [6].

Furthermore, in order to advance the development of alginate beads, as delivery systems for water-soluble drugs and proteins administered orally, it is crucial to investigate the swelling behavior as the primary parameter. The swelling profiles of calcium alginate beads have been thoroughly examined and analyzed in the existing literature using media that imitate the physiological conditions of the gastrointestinal tract. At pH 1.2, which is similar to the stomach’s pH, there is a slight swelling, while at pH 6.8, which is similar to the small intestine’s pH, there is strong and extensive swelling. Alginate hydrogels are effective as carriers for the oral administration of chemotherapeutics, water-insoluble medications, and proteins/peptides because of this pH-responsive behavior. Other materials, such as Pluronics, can be employed in the preparation of hydrogels to alter their swelling properties [1,2]. Additionally, the drug-loading and release profile of active substances can also be modified by the presence of triblock copolymers [1,2].

Pluronics, a group of block copolymers with both hydrophilic and hydrophobic components, have gained widespread use in medical and pharmaceutical applications, due to their remarkable biocompatibility, safety, and low immunogenicity. Pluronic F127, also named Poloxamer 407, is an FDA-approved temperature-sensitive hydrogel that has been widely used for drug delivery purposes and wound healing applications. The temperature-dependent gelation and micellization properties of these polymers also make them ideal as matrices and carriers, allowing for the efficient transport of active pharmaceutical ingredients (APIs), particularly those that are sparingly soluble in water [7]. Pluronic belongs to a family of amphiphilic block copolymers, which can self-assemble, forming nanosized micelles at the critical micelle concentration (CMC). The suitable range for the development of micelle systems is 0.04–0.46 mM [8,9]. Micelles usually have hydrophilic chains and hydrophobic cores. It has been observed that the presence of Pluronic^®^ F-127 in the polymeric network reduces the swelling of the hydrogel beads and this might be attributed to the hydrophobic nature of micelles leading to minimum swelling [8].

Similar methods have already been investigated, although disparities exist in the medicinal product integrated within the beads, the concentration of CaCl_2_, or the method of bead formation and subsequent drying. For instance, a combination of drugs, theophylline, and theobromine, using CaCl_2_ 0.27 M for the formation of beads, has been already used with the aid of a Braun Mekungen infusion pump [10]. Alternatively, the drug is dissolved in HCl, which is subsequently added to the polymeric solution, and the beads are formed through the use of a 2G syringe needle, under stirring [11].

The drug that was used in this study is acetylsalicylic acid (ASA). ASA or aspirin belongs to the family of salicylates and is used as an anti-inflammatory agent. Yet, nowadays it is also used as an anticoagulant drug against heart attacks and cancer. ASA is administered mainly orally and its absorption depends on many factors, such as the dosage form, the gastric emptying time, and the gastric pH. ASA is hydrolyzed to salicylic acid in the plasma and salicylate anion is metabolized in the liver. When it is delivered orally, it is absorbed both in the stomach and in the small intestine. Through passive diffusion, the non-ionized acetylsalicylic acid passes through the stomach, whilst its ionized form is absorbed ideally at pH = 2.15–4.1. However, the absorption of ASA takes place at a greater rate at the pH of the intestine [10]. ASA works by inhibiting the production of prostaglandins, which normally induce inflammation and pain. Aspirin is rapidly absorbed in the stomach and the upper small intestine, where it is hydrolyzed to salicylic acid, the active form of the drug. Salicylic acid is then absorbed into the bloodstream, where it circulates and exerts its effects on various tissues in the body.

However, aspirin can also cause irritation and damage to the mucosa of the stomach and intestine, leading to gastrointestinal bleeding and ulcers. To mitigate this, enteric-coated aspirin has been developed, which passes through the stomach and then dissolves in the small intestine, reducing the risk of gastric irritation [12].

The scope of this investigation is to compare the physicochemical behavior of low- and medium-alginate capacities as delivery platforms with pH-responsive release properties (Figure 1). For this reason, we examine the swelling ratios and the release profiles of two different viscosity types of alginate beads, low and medium viscosity, with and without Pluronic^®^ F-127, in order to evaluate their characteristics and release properties.

## 2. Materials and Methods

### 2.1. Materials

Low-viscosity (250 cps of 2% solution) and medium-viscosity (>2000 cps of 2% solution) alginic acid sodium salt (NaAlg) salt from brown algae, composed of *D*-mannuronic and *L*-guluronic acid chains and calcium chloride dihydrate (CaCl_2_**·**2H_2_O), as well as Pluronic^®^ F-127 (in the form of white microbeads) were purchased from Sigma-Aldrich (Athens, Greece) and used as received. The simulated gastric (SGF, pH 1.2) and the simulated intestinal fluids (SIF, pH 6.8) were prepared as described in the US Pharmacopeia.

### 2.2. Methods

#### 2.2.1. Preparation and Physicochemical Characterization of Micelles

The appropriate amounts of Pluronic^®^ F-127 were dissolved in purified water. The colloidal concentration of Pluronic^®^ F-127 is 100 mg/mL, 10% *w*/*v*. The dilution took place at room temperature under mild agitation. The size, size distribution, and zeta potential (ζ-potential) of the micelles were evaluated by dynamic and electrophoretic light-scattering techniques. Measurements were performed at a detection angle of 90° and at 25 °C in a photon correlation spectrometer (SZ-100, Horiba Ltd., Kyoto, Japan, 601–8510, Horiba ABX SAS, Montpellier, France) and analyzed by the CONTIN method (Horiba software 2009–2018). For each system, three sets of three light-scattering measurements were collected and the results were averaged. Specifically, 100 μL of dispersion was diluted to 3 mL of HPLC-grade water, and the zeta potential was measured at room temperature at 633 nm. The zeta-potential values were calculated from electrophoretic mobilities, *μ*_E_, by using the Henry correction of the Smoluchowski equation:$$\zeta =\frac{3\mu_{E}n}{2\varepsilon_{0}\varepsilon_{r}}\frac{1}{f\left(ka\right)}$$
where *ε*_0_ is the permittivity of the vacuum, *ε_r_* is the relative permittivity, *a* is the particle radius, *k* is the Debye length, and *n* is the viscosity of HPLC-grade water. All the experiments were run in triplicate. The zeta potential of the micelles was measured at the pH of HPLC-grade water, which is 5.5 at 25 °C.

#### 2.2.2. Preparation of sodium-alginate beads

The protocol used has been previously described in the literature [2]. Sodium alginate solutions 4% *w*/*v* of low (LV) and medium viscosity (LV) were prepared in order to synthesize the beads. The encapsulation of Pluronic^®^ F-127 micelles was achieved by mixing the sodium alginate solution with the Pluronic^®^ F-127 solution in a 9:1 weight ratio. Alginate solution droplets were introduced into a 0.1 M CaCl_2_ solution using a syringe pump, resulting in the formation of separate alginate beads for each alginate, all with the same flow rate. The beads were then left for 30 min in the CaCl_2_ 0.1 M solution in order to be properly formed. Last, the beads were rinsed with water to remove the excess CaCl_2_ and were dried for 24 h at 80 °C.

#### 2.2.3. Thermogravimetric Analysis

Thermogravimetric analysis (TGA) was performed with a TA Instruments Q500 TGA Analyzer in the temperature range of 30–900 °C at a heating rate of 10 °C/min under air atmosphere.

#### 2.2.4. Morphological Characterization

The morphological characterization of the different alginate beads was conducted using a Phenom-World desktop scanning electron microscope (SEM) (Phenom-World, Eindhoven, The Netherlands) with a tungsten filament operating at 10 kV and a charge reduction sample holder. The samples were observed after sputter coating on a Mini Sputter Coater/Glow Discharge System SC7620 (Quorum Technologies Ltd., Lewes, UK).

#### 2.2.5. Swelling Studies

The swelling studies took place in two different dispersion media, in accordance with the reported protocols [2,13].

This is carried out by weighing the pellets at time *t* = 0, being in the dry state after 48 h in the dryer and then in regular time intervals at the different pH solutions, pH = 1.2 (SGF medium), and then at pH = 6.8 (SIF medium). Accurately weighed amounts of beads (ranging from 0.7 to 1.2 mg) were immersed in a 25 mL SGF solution for two hours and then transferred into a 25 mL SIF solution (pH 6.8) for four more hours. At fixed time intervals, the beads were separated from the medium using a stainless-steel grid and were immediately wiped gently with paper and weighed. The %weight change of the beads with respect to time was calculated according to the following mathematical formula and is called swelling ratio *Q_s_*:(1)$$Q_{s}=\frac{\left|W_{d}-W_{s}\right|}{W_{d}}\times 100\%$$
where, *W_s_* is the weight of the beads in the swollen state and *W_d_* is the initial weight of the dry beads. 

#### 2.2.6. Dissolution Studies

The dissolution studies were carried out using the dissolution bath method. The motor was set at 50 rpm and the constant temperature bath at 37 °C. Two simulated fluids of 1000 mL each were prepared to evaluate the drug release performance of the LV and MV alginate beads. Three out of the six vessels were filled with 300 mL of the SGF medium (sodium chloride buffer at pH = 1.2), while the SIF medium (pH = 6.8) was added in the remaining three [13]. The alginate beads were placed at the dissolution baskets with the simulated gastric fluid for 2 h and then transferred in the intestinal fluid. From each dissolution basket, 5 mL was withdrawn at specific time points and the volume of the medium was replenished by adding the same amount of each buffer. The drug release of each sample was determined by measuring the absorbance of acetylsalicylic acid, using UV–Vis spectrophotometry, at 276 and 296 nm, for the samples at pH = 1.2 and 6.8, respectively. 

In order to compare the dissolution profiles of the LV and MV alginate beads, graphs of % acetylsalicylic acid release vs. time were constructed. The dissolution efficiency (DE) is equal to the area under the dissolution curve between the time points, *t*_1_ and *t*_2_, and is expressed as the percentage of the curve at maximum dissolution, y_100_, at the same time period (Equation (2)) [14]:(2)$$\mathrm{DE}=\frac{\int_{t_{1}}^{t_{2}}y\times dt}{y_{100}\times (t_{2}-t_{1})}\times 100$$
where, y is the percentage of the dissolved product.

The sampling time, *t*_x%_, corresponds to the time that x% of the active substance has been released from the pharmaceutical formulation and therefore, the values *t*_20%_, *t*_50%_, and *t*_90%_ were calculated.

The mean dissolution time (*MDT*) values were calculated from the following equation:(3)$$MDT=\frac{ABC}{W_{\infty }}$$
where, $W_{\infty }$ is the maximum amount of acetylsalicylic acid dissolved, and *ABC* is the area between the drug dissolution curve and its asymptote [14].

In order to determine the kinetics of dissolution, the in vitro release data were modeled using the Korsmeyer–Peppas equation:(4)$$\frac{M_{t}}{M_{\infty }}=kt^{n}$$

The variables $M_{t}$ and $M_{\infty }$ represent the total amount of drug released at a specific time and infinite time, respectively. The release rate constant is denoted by *k*, while *n* represents the diffusion coefficient. For cylindrical tablets, when *n* is less than or equal to 0.45, the release follows Fickian diffusion kinetics (case I diffusional). If the *n* value is between 0.45 and 0.89, the release is non-Fickian with anomalous transport. When *n* is equal to 0.89, the release follows zero-order (case II) kinetics.

In order to compare the dissolution curves, the difference factor (*f*_1_) and the similarity factor (*f*_2_) are used. Factors *f*_1_ and *f*_2_ presented by the Equations (5) and (6), respectively, are qualitative (model independent) approaches to compare the dissolution pairs. The *f*_1_ factor measures the average relative difference between two curves across all the time points, whereas the *f*_2_ factor is used in order to compare the cumulative curves of percentage dissolution between two samples [15]:(5)$$f_{1}=\left\{\frac{\left[\sum_{t=1}^{n}\left|R_{t}-T_{t}\right|\right]}{\left[\sum_{t=1}^{n}R_{t}\right]}\right\}\times 100$$
(6)$$f_{2}=50\times \log\left\{{\left[1+\frac{1}{n}\sum_{t-1}^{n}{\left(R_{t}-T_{t}\right)}^{2}\right]}^{-0.5}\times 100\right\}$$
where, *n*, is the number of time points; *R_t_*, mean % reference drug dissolved at time *t* after initiation of the study; *T_t_*, mean % test drug dissolved at time *t* after initiation of the study. 

The *f*_2_ value ranges from 0 to 100, with the absolute similarity having the value of 100. As the difference between the test and reference profiles increases, the *f*_2_ score decreases and can approach zero. However, due to *f*_2_’s logarithmic nature, even minor differences between the profiles can lead to a significant decrease in the *f*_2_ score. The two dissolution curves are similar when the *f*_2_ value lies between 50 and 100 and the *f*_1_ value is between 0 and 15.

#### 2.2.7. Statistical Analysis

Results are shown as the mean value-standard deviation (SD) of three independent measurements. Statistical analysis was performed using the Student’s *t*-test, and multiple comparisons were carried out using one-way ANOVA; *p*-values < 0.05 were considered statistically significant. All statistical analyses were performed using “Excel 2016” (Microsoft, Redmond, WA, USA).

## 3. Results

### 3.1. Physicochemical and Thermotropic Characterization of the Prepared Systems

Pluronic^®^ F-127 is self-assembled into micelles with a size around 120 nm (Table 1) with a homogenous population of the nanoparticles (Table 1), considering the mild protocol of the preparation, without any method of size reduction. The ASA caused a slight decrease in the particle size of the Pluronic^®^ F-127 micelles. The Pluronic^®^ F-127:ASA nanoparticles exhibited a 100 nm size (Table 2) with a higher degree of homogeneity of nanoparticulate population (Table 2). In both cases, the ζ-potential values are close to zero, indicating the absence of surface network charge on the colloidal particles (Table 2). These ζ-potential values of the Pluronic^®^ F127 micelles with the encapsulated API are similar to those recorded in the literature. Additionally, according to the published data, the ζ-potential values did not alter, irrespective of the pH of the medium (ranging from 5 to 7). This suggests that the surface characteristics of the prepared micelles did not change in conditions mimicking the gastrointestinal tract [16].

We also checked the colloidal stability of the Pluronic^®^ F-127 nanomicelles with and without the encapsulated ASA for a period of 30 days. Both dispersions were stored at 4 °C (in the fridge). The results from the physicochemical characterization after 30 days showed that both the nanomicellar formulations are stable (Dh ≈ 140 nm for pure micelles and Dh ≈ 125 nm for ASA micelles; the PDI values are slightly increased, and the ζ-potential remained unaffected). The last observation suggests that steric stabilization due to the hydrophilic chains of the triblock copolymer is responsible for keeping the micelles in dispersion state, without the presence of electrostatic repulsion.

TGA was performed in order to evaluate the thermal stability of the hydrogel beads. The TGA curve (Figure 1) shows the % weight loss in relation to the increase in temperature. As the temperature increases, the weight decreases, and the mass changes due to thermal treatment.

According to Figure 1, the initial weight loss occurs in the range from 30 °C to 175 °C and about 10% of the weight is lost during this temperature range, due to the loss of trapped water. The loss of water is due to the hydrophilic nature of hydrogels, resulting from the presence of -COOH, -OH groups [8,17,18]. The major weight loss is observed at the two degradation processes that occur in the regions of 200 °C to 250 °C and 250 °C to 300 °C, probably due to alginate backbone destruction and hydroxyl group loss [2].

Further heating showed two more degradation processes of mass loss in the range from 300 °C to 450 °C and occurring due to the decomposition of oxygen-containing groups, such as hydroxyl and carboxyl groups (COO-) releasing CO_2_ and methane and water being formed [19,20]. Above 500 °C, the mass loss rate is slower due to the egg-box structure that the crosslinked beads have [21]. The total weight loss for all beads is about 60%. The beads with Pluronic do not seem to exhibit different behavior. This indicates that Pluronic^®^ F-127 does not award any thermal stability to the treated sodium alginate beads [22].

Although we would expect beads with medium viscosity to lose more weight, medium- and low-viscosity beads in this TGA demonstrated similar behavior. According to the published literature, MV beads are larger in size compared to LV beads. Therefore, they can store more water inside, and thus, they should lose more mass, especially in the first stage of TGA, where the mass is lost due to the evaporation of moisture [22].

### 3.2. Swelling Studies

The swelling profile of the dry beads (Figure 2) in SGF (2 h), and in SIF (4 h) is demonstrated in Figure 3. The swelling of the beads is presented in the graph as the mass change (%) vs. time (min) [18]. 

First, we notice that the viscosity of the sodium alginate affected the size and shape of the droplets formed. When a higher viscosity (MV) sodium alginate was used, the resulting beads were larger than those prepared with a lower viscosity (LV) alginate. Additionally, the beads made with the higher viscosity alginate had a higher capacity to swell [22]. The swelling was found to be dependent on the viscosity of sodium alginate, since medium-viscosity (MV) alginate beads swell to a greater extent than their low-viscosity (LV) counterparts. The increase in the LV beads’ mass reaches 1309% at 4 h, where the MV beads swell up to 2920%. This could be attributed to the varying number of monomer units present in the crosslinked chains of alginate [17]. The viscosity at similar concentrations is directly proportional with the size of chains. Therefore, water absorption in the MV beads rises due to the increased endurance of the hydrogels [18]. The effects of pH ranging from 1 to 10 on the morphology and shape of alginate beads with different concentrations of alginate solutions have already been reported in the literature [23,24]. In these studies, the alginate beads exhibited had a more spherical structure at neutral pH [23,24]. In the present investigation, we focused on how the presence of Pluronic^®^ F127 micelles changes the swelling behavior of the alginate beads with different degrees of viscosity. For this reason, we only used two buffer solutions mimicking the conditions of the gastrointestinal tract.

Furthermore, as demonstrated in Figure 3, the alginate beads with Pluronic^®^ F-127 dissolve faster at pH = 6.8 with the MV alginate beads dissolving in 30 min and the LV alginate beads in 120 min, at pH = 6.8. When the pH increases, the ionized core is enhanced by repulsive interactions that lead to the increase in the micellar size, until they dissolve [7]. However, we noticed that, once again, the MV alginate beads had better swelling capacity and lasted for a longer period, at pH = 6.8. 

In addition, the aqueous medium also has an effect on the swelling of the beads. As shown in Figure 3, the developed hydrogel beads indicated a pH-dependent swelling behavior. In the SGF medium (pH = 1.2), the swelling is negligible in comparison to the SIF medium (pH = 6.8), where the swelling is significantly higher, with and without Pluronic^®^ F-127. This can be explained by the fact that the alginate amphiphilic micelles bear ionized carboxylic acid groups. At pH = 1.2, these groups are protonated to -COOH [25]. Therefore, the swelling ratios of the alginate beads are decreased at pH = 1.2 in comparison with those at pH = 6.8, due to the enhanced ionization of the carboxylic acid groups in solutions of higher pH. When the sodium alginate beads are present in an alkaline environment, the Ca^2+^ ions are exchanged with K^+^. The Ca^2+^ ions are binding with the carboxylic acid groups, and this leads to decreased electrostatic repulsions causing chain loosening and enhancement of the gel swelling. Intermolecular cross-linkage is being formed between Ca^2+^ and G Blocks, shaping a hard surface of the beads [26,27]. However, after some hours, dissolution of the beads takes place, since the Ca^2+^ ions are gradually released to the medium [26].

### 3.3. Morphological Characterization

SEM (scanning electron microscopy) provides information about the surface morphology and microstructure of hydrogels. SEM, as shown in Figure 4, can reveal structural differences between the LV and MV hydrogels, with and without Pluronic^®^ F-127.

The surface of hydrogels without Pluronic^®^ F-127 appears porous and voids or interconnected pores are evident. However, there is a difference between the low- and medium-viscosity hydrogels. The surface of the LV hydrogels is smoother in contrast to that of the MVs, which are rougher with wrinkles (Figure 4a,c). This is because higher viscosity limits the mobility of polymer chains leading to denser morphology, while low viscosity allows for an easier spread of hydrogel (Figure 4a,c).

When characterizing hydrogels with Pluronic^®^ F-127, several observations can be made. First, Pluronic^®^ F-127 can self-assemble into spherical structures, micelles. These micelles are distributed throughout the hydrogel depending on the concentration of Pluronic^®^ F-127. Medium-viscosity hydrogels with Pluronic^®^ F-127 appear with a structured network exhibiting a variety of pore sizes and a moderate level of porosity (Figure 4b,d). Higher viscosity leads to the formation of smaller and more interconnected pores compared to the LV hydrogels, which have larger and more loosely distributed pores (Figure 4c,d). Generally, the MV hydrogels undergo a controlled gelation process exhibiting a more stable structure compared to the LV hydrogels. Higher viscosity leads to an efficient crosslinking of the polymeric chains, creating a robust and interconnected network [28].

### 3.4. Dissolution Studies

The % in vitro drug release of low- and medium-viscosity alginate beads with Pluronic^®^ F-127 results are presented in Figure 5. The triblock copolymer is used for the dissolution of the API and the micellar API’s encapsulation in the alginate beads.

As displayed in Figure 5, the drug release of aspirin from both the LV and MV alginate beads differs at pH = 1.2 and 6.8. In the acidic environment (pH = 1.2), the drug release rate of both the LV and MV alginate beads is reduced. Consequently, the release of aspirin from the beads occurs at a slower pace. Conversely, when the pH is 6.8, both the LV and MV alginate beads exhibit an increased drug release rate. At pH = 1.2, the carboxylate anions of the alginate backbone are protonated to -COOH and the electrostatic interactions are decreased. Regarding the interactions between the drug and the polymer, Pluronic^®^ F-127 increases the drug bioavailability by prolonging the circulation of the drug inside the beads [7]. The LV alginate beads have smaller carboxylated chains and, therefore, they can dissolve easier. At 2 h, the release of LV drug-loaded beads reaches 63% in the gastric fluid (pH = 1.2), whereas it reaches 30% for the MV drug-loaded beads. At pH = 6.8, rapid drug release occurs, where in only 1 h, 40% and 60% of the drug is released from the LV and MV alginate beads, respectively. The facile release of acetylsalicylic acid, at the alkaline medium, is possibly due to the increased electrostatic interactions between Ca^2+^ and the ionized carboxylic acid groups, which leads to a 100% release of the drug from both the LV and MV alginate beads. Additionally, the p*K*a value contributes significantly to the accelerated release of aspirin. The p*K*a value of acetylsalicylic acid, which is considered a weak acid, has been reported to range from 3.48 to 3.5 to 3.6, depending on the supplier. As a result, the solubility of ASA is pH dependent, with solubility increasing as the pH surpasses the p*K*a value, at pH = 6.8, in this case [29]. This result satisfies our target, as the acetylsalicylic acid was sought to be dissolved in the small intestine to avoid side effects on the stomach but also to have a quick effect. “We performed the experiments for 180 min because during this time period all the encapsulated ASA was released. In our opinion, this release profile is a combination of the pH-responsive behavior of alginate beads and the long-term release of the Pluronic^®^ F-127 nanomicelles. The nanomicelles act as a “second” barrier that should be destroyed for the release of the encapsulated ASA.

The kinetic release properties of the developed formulations are reported in Table 3.

The values *t*_20%_, *t*_50%_, and *t*_90%_ are higher for the MV alginate beads, due to their longer chains [18]. Therefore, they have a higher mechanical strength, which is enhanced due to the presence of Pluronic^®^ F-127 [30]. Furthermore, based on Table 2, 20% of the drug was released in the acidic medium from both the LV and MV alginate beads. However, 50% of acetylsalicylic acid is released from the LV beads in the acidic medium, where 50% of the released drug in the MV beads occurs in the alkaline medium, which differentiates the two formulations according to the desired application. Furthermore, in this experiment, the values of *f*_1_ = 34.03 and *f*_2_ = 39.92 indicate that the dissolution curves are not similar (Table 2).

The dissolution kinetics of the alginate beads can be understood through the Korsmeyer–Peppas model. Thus, the LV alginate beads have an *n* value of 0.57, indicating that their drug release mechanism follows Fickian diffusion (*n* = 0.5) and zero-order release (*n* = 1.0), characterized as anomalous release. On the other hand, the MV alginate beads have an *n* value of 0.4, which suggests a release mechanism closer to a non-Fickian or anomalous release (0 < *n* < 0.5) [14].

## 4. Conclusions

In this investigation, we designed and developed alginate beads for the encapsulation and release of ASA. To the best of the authors’ knowledge, this is the first report, where a comparison study was designed to evaluate the physicochemical and release behavior of beads composed of different viscosity alginates. In Table 3, the differences between these excipients are summarized. Low-viscosity hydrogels are typically formulated with a higher water concentration, leading to a less compact gel structure. This formulation enables easier handling and encapsulation of bioactive substances, resulting in faster release kinetics. Moreover, the low-viscosity hydrogels possess a more porous structure, allowing for a greater degree of permeability. On the other hand, the medium-viscosity hydrogels have a denser and more uniform gel structure, leading to lower permeability. This structural characteristic gives rise to slower release kinetics. Regarding their thermotropic profile, both types of hydrogels exhibit more or less the same behavior. In terms of the swelling behavior, the medium-viscosity hydrogels demonstrate a moderate degree of swelling, which contributes to the mechanical integrity and long-term stability of the gel. In conclusion, this comparative study could be used as a roadmap for formulation scientists for the selection of biomaterials/multifunctional excipients in order to design and develop delivery platforms with the desired release kinetic profile. 

## Data Availability

Not applicable.

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
