# Peer review of "Design and Development of Low- and Medium-Viscosity Alginate Beads Loaded with Pluronic^®^ F-127 Nanomicelles"

_materials, 2023, doi:10.3390/ma16134715_

Round 1
Reviewer 1 Report
This manuscript describes the swelling ratios of low and medium viscosity alginate beads with and without Pluronic® F-127 to evaluate their properties and in vitro drug release characteristics.
Major concerns:
1- To better illustrate the design study, a schematic representation showing the synthesis of the hydrogel beads, the network structure, the drug uptake and the release profile is required.
2- Figure 1 shows the TGA thermograms of the alginate beads under different conditions, showing the effect of temperature on weight loss. A wide range of temperatures was chosen. It would be nice to see the SEM or images of the hydrogel beads under different temperature conditions. Water loss causes shrinkage of the hydrogel beads, and I think a profile of the effect of temperature on the size of the beads should be provided. I propose to make a schematic representation of Alginate MV, Alginate MV+Pluronic, LV-Alginate and LV-Alginate+Pluronic in a panel figure and combine it with Figure 1. This way, the figure looks better and is easier to illustrate to the readers.
3- How do the authors measure zeta potential and how many samples were used? Please explain in detail. At what pH values are the zeta potential measured? I would suggest measuring the zeta potential at different acidic and basic conditions and plot a curve for better understanding. By adjusting the pH, one can influence the behavior of the polymers, which is one of the most important parameters for thermo-responsive polymers.
4- The captions of all figures need improvement and require a detailed explanation with conditions. For example, in Figure 3, at what time and pH were the images taken? How much was the size increased? There are many studies in the literature showing the effect of pH on the swelling of hydrogel beads;
https://doi.org/10.1155/2017/3902704
https://doi.org/10.1002/mabi.200600013
Quantitative data is needed for this study to determine the swelling and shrinking ratio at different pH values. In addition, Figure 3 shows that the size increases by hundreds to thousands of times, and no size distribution curve is provided. In drug delivery, monodispersity and temporal stability are the most important parameters. Therefore, authors should provide both size distribution and stability curves.
5- Figure 6 shows SEM micrographs of low and medium viscosity alginate beads with Pluronic. I would like to see the morphology of the beads, spherical or semi-spherical at different pH values. Please explain the experimental conditions in the caption.
6- There are existing studies on similar topics and contents
DOI: 10.1208/s12249-010-9406-z
https://doi.org/10.1016/j.indcrop.2022.116081
To support the publication, it is necessary to specify the new addition with justification.
7. Sentence structures need to be improved throughout the manuscript, there are examples in the minor comments section. It was really hard for me to understand what the authors were trying to convey.
Minor comments:
1- I would recommend rephrasing the following sentences;
Line 39-42 “due to their biocompatibility and biodegradability, natural polysaccharides are being explored for potential use in pharmaceutical and food industries. One such polysaccharide is sodium alginate (SA), which is commonly used in pharmaceutical formulations and drug delivery systems”.
- Line 45, Multifunctional hydrogels have numerous applications in biomedical sciences. Please name a few hydrogels with references.
- Line 57: One such polymer, SA, has a high capacity to chelate divalent cations.
-Lines 61-62: Due to its pH-sensitivity and exceptional ability to form gels, SA has become a 61 popularly researched and developed material for drug delivery systems meant for oral 62 administration.
- Lines 70-71: This pH-responsive 70 behavior is responsible.
- Line 96: using CaCl2 0.27M and the beads were formed
- Line 102-103: Prove, via dissolution 102 studies, that alginate beads with ASA are dissolved at the SIF medium.
2- Provide the reference for the pH values of the stomachs and small intestines. By the way, this is a dynamic range and not a fixed value.
3- Define the zeta potential acronym in the Materials and Methods section.
4- What is meant by a mild protocol?
5- In the section 3.1, please change the way the temperature range is indicated, e.g., 30 to 175 °C.
Substantial revisions are required.
Author Response
Major concerns:
1- To better illustrate the design study, a schematic representation showing the synthesis of the hydrogel beads, the network structure, the drug uptake and the release profile is required.
In response to the reviewer’s comment, a schematic representation of the formulations of this study was added.
Scheme 1. Schematic representation of: A. Pluronic® F127 micelles, B. network of alginate beads, C. the formation of alginate beads, D. Pluronic® F127 micelles loaded with ASA and E. alginate beads encapsulated with Pluronic® F127 micelles loaded with ASA.
2- Figure 1 shows the TGA thermograms of the alginate beads under different conditions, showing the effect of temperature on weight loss. A wide range of temperatures was chosen. It would be nice to see the SEM or images of the hydrogel beads under different temperature conditions. Water loss causes shrinkage of the hydrogel beads, and I think a profile of the effect of temperature on the size of the beads should be provided. I propose to make a schematic representation of Alginate MV, Alginate MV+Pluronic, LV-Alginate and LV-Alginate+Pluronic in a panel figure and combine it with Figure 1. This way, the figure looks better and is easier to illustrate to the readers.
Figure 1 shows TGA thermograms of the prepared alginate beads of low and medium viscosity with and without the encapsulation of Pluronic® F127 micelles. For this experiment, we did not use different conditions, i.e., pH. This is a comparison study in order to evaluate the physicochemical and release behavior of beads composed of different viscosity alginates. As mentioned in the manuscript, there are other reports in the literature studying the behavior of alginate beads under different pH conditions.
3- How do the authors measure zeta potential and how many samples were used? Please explain in detail. At what pH values are the zeta potential measured? I would suggest measuring the zeta potential at different acidic and basic conditions and plot a curve for better understanding. By adjusting the pH, one can influence the behavior of the polymers, which is one of the most important parameters for thermo-responsive polymers.
According to the reviewer’s comment, in the section “2.2.1. Preparation and physicochemical characterization of micelles”, we added: “Measurements were performed at a detection angle of 90° and at 25 °C in a photon correlation spectrometer (SZ-100, Horiba Ltd, Kyoto, 601-8510, Horiba ABX SAS, Montpellier, France) and analyzed by the CONTIN method (Horiba software). For each system, three sets of three light scattering measurements were collected and the results were averaged. Specifically, 100 μL of dispersion was diluted to 3 mL of HPLC grade water, and the zeta-potential was measured at room temperature at 633 nm. The zeta-potential values were calculated from electrophoretic mobilities, μE, by using the Henry correction of the Smoluchowski equation:
where, ε0 is the permittivity of the vacuum, εr is the relative permittivity, a is the particle radius, k is the Debye length, and n is the viscosity of HPLC-grade water. All the experiments were run in triplicate. The zeta potential of the micelles was measured at the pH of HPLC-grade water, which is 5.5 at 25°C).”
In the section “3.1 Physicochemical and thermotropic characterization of the prepared systems” line 235, we added: “These ζ-potential values of the Pluronic® F127 micelles with the encapsulated API are similar to those recorded in the literature. Additionally, according to the published data, the ζ-potential values did not alter, irrespectively of the pH of the medium (ranging from 5 to 7). This suggests that the surface characteristics of the prepared micelles did not change in conditions mimicking the gastrointestinal tract. (Shaarani et at., 2017)”.
In the reference list, we also added: “Shaarani S, Hamid SS, Mohd Kaus NH. The Influence of pluronic F68 and F127 nanocarrier on physicochemical properties, in vitro release, and antiproliferative activity of thymoquinone drug. Phcog Res 2017;9:12-20.doi: 10.4103/0974-8490.199774.”
4- The captions of all figures need improvement and require a detailed explanation with conditions. For example, in Figure 3, at what time and pH were the images taken? How much was the size increased? There are many studies in the literature showing the effect of pH on the swelling of hydrogel beads;
https://doi.org/10.1155/2017/3902704
https://doi.org/10.1002/mabi.200600013
Quantitative data is needed for this study to determine the swelling and shrinking ratio at different pH values. In addition, Figure 3 shows that the size increases by hundreds to thousands of times, and no size distribution curve is provided. In drug delivery, monodispersity and temporal stability are the most important parameters. Therefore, authors should provide both size distribution and stability curves.
We changed the order of Figures 2 and 3 to avoid misunderstanding. According to the reviewer’s comment, we changed the caption of Figure 2 as follows: “The hydrogel beads after the drying process for 24h at 80°C (left) and after their formation in the CaCl2 0.1M solution (right)”
In the section “3.3. Swelling studies”, we added: “The effects of pH ranging from 1 to 10 on the morphology and shape of alginate beads with different concentrations of alginate solutions have already been reported in the literature (Shi et al., 2006; Chuang et al., 2017). In these studies, the alginate beads had a more spherical structure at neutral pH (Shi et al., 2006; Chuang et al., 2017). In this investigation, we focused on how the presence of Pluronic® F127 micelles changes the swelling behavior of the alginate beads with different degrees of viscosity. For this reason, we only used two buffers mimicking the conditions of the gastrointestinal tract.”
In the reference list we also added:
Jui-Jung Chuang, Yu-Ya Huang, Szu-Hsuan Lo, Tzu-Fang Hsu, Wen-Ying Huang, Shu-Ling Huang, Yung-Sheng Lin, "Effects of pH on the Shape of Alginate Particles and Its Release Behavior", International Journal of Polymer Science, vol. 2017, Article ID 3902704, 9 pages, 2017. https://doi.org/10.1155/2017/3902704
Shi J, Alves NM, Mano JF. Drug release of pH/temperature-responsive calcium alginate/poly(N-isopropylacrylamide) semi-IPN beads. Macromol Biosci. 2006 May 23;6(5):358-63. doi: 10.1002/mabi.200600013.
5- Figure 6 shows SEM micrographs of low and medium viscosity alginate beads with Pluronic. I would like to see the morphology of the beads, spherical or semi-spherical at different pH values. Please explain the experimental conditions in the caption.
In the section “3.3. Swelling studies”, we added: “The effects of pH ranging from 1 to 10 on the morphology and shape of alginate beads with different concentrations of alginate solutions have already been reported in the literature (Shi et al., 2006; Chuang et al., 2017). In these studies, the alginate beads had a more spherical structure at neutral pH (Shi et al., 2006; Chuang et al., 2017). In this investigation, we focused on how the presence of Pluronic® F127 micelles changes the swelling behavior of the alginate beads with different degrees of viscosity. For this reason, we only used two buffers mimicking the conditions of the gastrointestinal tract.”
According to the reviewer’s comment, we changed the caption of Figure 4, as follows: Figure 4. SEM micrographs [scale bar = 20 μm (´4256)] of a) LV alginate beads, b) LV alginate beads with Pluronic® F-127, c) MV alginate beads and d) MV alginate beads with Pluronic® F-127. Dry samples were observed after sputter coating.
6- There are existing studies on similar topics and contents
DOI: 10.1208/s12249-010-9406-z
https://doi.org/10.1016/j.indcrop.2022.116081
To support the publication, it is necessary to specify the new addition with justification.
According to the reviewer’s comment, we added the above references in the Introduction part and we provided justification of the scope of this study:
“Alginate, calcium chloride, and chitosan were ionotropically gelled to create bovine serum albumin-loaded beads. Particle size and encapsulation effectiveness of chitosan-alginate beads were strongly influenced by the chitosan content. At the higher pH, the chitosan-alginate matrices quickly dissolved, causing a burst release of the protein (Takka et al., 2010). According to the recent literature, an acetic acid coagulation bath and Ca2+ chelation were used to make nanocellulose and sodium alginate miscible, creating an interpenetrating network hydrogel via double crosslinking with considerable prolongation of the aspirin release period to around 60 hours (Ma et al., 2023).”
In the reference list, we also added:
Takka, S., Gürel, A. Evaluation of Chitosan/Alginate Beads Using Experimental Design: Formulation and In Vitro Characterization. AAPS PharmSciTech 11, 460–466 (2010). https://doi.org/10.1208/s12249-010-9406-z.
Huazhong Ma, Jianglin Zhao, Ying Liu, Liang Liu, Juan Yu, Yimin Fan. Controlled delivery of aspirin from nanocellulose-sodium alginate interpenetrating network hydrogels. Industrial Crops and Products, 192, 2023, 116081.
- Sentence structures need to be improved throughout the manuscript, there are examples in the minor comments section. It was really hard for me to understand what the authors were trying to convey.
We made significant improvements in the manuscript’s text in order to avoid misunderstandings.
Minor comments:
1- I would recommend rephrasing the following sentences;
Line 39-42 “due to their biocompatibility and biodegradability, natural polysaccharides are being explored for potential use in pharmaceutical and food industries. One such polysaccharide is sodium alginate (SA), which is commonly used in pharmaceutical formulations and drug delivery systems”.
- Line 45, Multifunctional hydrogels have numerous applications in biomedical sciences. Please name a few hydrogels with references.
“Multifunctional hydrogels based on alginates, chitosan, ulvan and other polysaccharides, have numerous applications in biomedical sciences, including drug delivery, wound dressing, tissue engineering, and temporary scaffolding for cells.”
- Line 57: One such polymer, SA, has a high capacity to chelate divalent cations.
-Lines 61-62: Due to its pH-sensitivity and exceptional ability to form gels, SA has become a 61 popularly researched and developed material for drug delivery systems meant for oral 62 administration.
- Lines 70-71: This pH-responsive 70 behavior is responsible.
- Line 96: using CaCl2 0.27M and the beads were formed
- Line 102-103: Prove, via dissolution 102 studies, that alginate beads with ASA are dissolved at the SIF medium.
We rephrased all the above sentences.
2- Provide the reference for the pH values of the stomachs and small intestines. By the way, this is a dynamic range and not a fixed value.
This point, raised by the Reviewer is correct. The pH variation along the gastrointestinal track is indeed a dynamic process. It depends on many factors, like the conversion of food into nutrients, which is primarily governed by the chemical reaction of hydrolysis, accompanied by a variety of enzymatically-controlled reactions, e.g. amylolysis, lipolysis, proteolysis, etc. Apart from temperature, these enzymatic processes depend on pH. The pH in the stomach may vary within the range from 6.5 (fed state) to 1.2 (fasted state), and in the small intestine, pH can range from 5.5 (duodenum) to 7.5 (ileum). Moreover, ionic strength and pH modulate the electrostatic charge, conformation, and solubility of molecules, including drug molecules.
Thus, it is scientifically sound the in vitro dissolution experiments to be conducted at two nodal pH buffer media, 1.2 and 6.8, in order to simulate, to a degree, the environment the per os administered drug substance is exposed to (vide: U.S. Department of Health and Human Services Food and Drug Administration Center for Drug Evaluation and Research (CDER) August 1997 - Guidance for Industry Dissolution Testing of Immediate Release
Solid Oral Dosage Forms:
https://www.fda.gov/regulatory-information/search-fda-guidance-documents/dissolution-testing-immediate-release-solid-oral-dosage-forms
We added the following reference in the section of dissolution studies:
“U.S. Department of Health and Human Services, Food and Drug Administration, Center for Drug Evaluation and Research (CDER). Guidance for industry: dissolution testing of immediate release solid oral dosage forms. Rockville: Food and Drug Administration, 1997 (available at:https://www.fda.gov/regulatory-information/search-fda-guidance-documents/dissolution-testing-immediate-release-solid-oral-dosage-f”
3- Define the zeta potential acronym in the Materials and Methods section.
We defined the zeta potential acronym in the Materials and Methods section as “ζ-potential”.
4- What is meant by a mild protocol?
We rephrased it as follows: “considering a preparation protocol without any additional method for size reduction”.
5- In the section 3.1, please change the way the temperature range is indicated, e.g., 30 to 175 °C.
We revised the temperature range according to the reviewer’s comment.

Reviewer 2 Report
1. Introduction: I recommend focus on alginate beans not on hydrogels; highlight the novelty of study; formulate goal not objectives.
2. Methods: different expressions of concentrations, it is more clear to use percentages. What statistics was used?
3. Results: figure 3 should be improved - photos of different formulations.
4. Discussion part with comparison to other scientific studies would increased the impact of this manuscript.
Author Response
- Introduction: I recommend focus on alginate beans not on hydrogels; highlight the novelty of study; formulate goal not objectives.
According to the reviewer’s comment, we changed last paragraph of the introduction, as follows: “The scope of this investigation is to compare the physicochemical behavior of low and medium alginate capacities as delivery platforms with pH-responsive release properties. For this reason, we examined the swelling ratios and the release profiles of two different viscosity types of alginate beads, low and medium viscosity, with and without Pluronic® F-127, in order to evaluate their characteristics and release properties.”
- Methods: different expressions of concentrations, it is more clear to use percentages. What statistics was used?
In the section 2.2.1 Preparation and physicochemical characterization of micelles, we added: “The colloidal concentration of Pluronic® F-127 is 100mg/ml, 10% w/v”.
We also added the section “2.2.7 Statistical analysis”
“Results are shown as the mean value-standard deviation (SD) of three independent measurements. Statistical analysis was performed using the Student’s t-test, and multiple comparisons were done using one-way ANOVA. P-values < 0.05 were considered statistically significant. All statistical analyses were performed using “Excel”.”
- Results: figure 3 should be improved - photos of different formulations.
We improved the figure 3 caption, and we added a new scheme, showing the different formulations.
- Discussion part with comparison to other scientific studies would increased the impact of this manuscript.
In the section “3.3. Swelling studies”, we added: “The effects of pH, ranging from 1 to 10, on the morphology and shape of alginate beads with different concentrations of alginate solutions have already been reported in the literature (Shi et al., 2006; Chuang et al., 2017). In these studies, the alginate beads exhibited a more spherical structure at neutral pH (Shi et al., 2006; Chuang et al., 2017). In the present investigation, we focused on how the presence of Pluronic® F127 micelles changes the swelling behavior of alginate beads with different degrees of viscosity. For this reason, we only used two buffer solutions mimicking the conditions of the gastrointestinal tract.”
In the reference list we also added:
Jui-Jung Chuang, Yu-Ya Huang, Szu-Hsuan Lo, Tzu-Fang Hsu, Wen-Ying Huang, Shu-Ling Huang, Yung-Sheng Lin, "Effects of pH on the Shape of Alginate Particles and Its Release Behavior", International Journal of Polymer Science, vol. 2017, Article ID 3902704, 9 pages, 2017. https://doi.org/10.1155/2017/3902704
Shi J, Alves NM, Mano JF. Drug release of pH/temperature-responsive calcium alginate/poly(N-isopropylacrylamide) semi-IPN beads. Macromol Biosci. 2006 May 23;6(5):358-63. doi: 10.1002/mabi.200600013.
Reviewer 3 Report
The manuscript by Kalogeropoulou et al. presented the design of a hydrogel-based drug carrier using alginate. The authors studied two main factors that influence the physicochemical properties of the polymer networks, including the viscosity (molecular weight) of the sodium alginate precursor and the introduction of Pluronic F127. Varying the two factors, the authors explored the swelling behaviors of the four samples, with additional characterizations under SEM to show their morphology differences. The authors further loaded a model drug in Pluronic containing alginate hydrogel beads and studied the drug delivery kinetics, suggesting the potential application of the materials as drug delivery platforms. The manuscript contains a full spectrum of characterizations to understand the materials, which helps to support that this is a promising material. However, there are also several noticeable flaws and improper claims in this manuscript. Therefore, I cannot recommend the publication of this manuscript before the following comments have been addressed.
(1) The chemical composition of the alginate beads needs to be examined. The current term “sodium-alginate beads” is inappropriate, considering that the sodium alginate solution undergoes ion exchange with Ca2+ chelating the carboxylates of alginate when dropped into CaCl2.
(2) The manuscript fails to mention the concentration of F127. The term “appropriate amounts of Pluronic...” is vague. The authors should provide quantitative details.
(3) The claim that alginate and F127 mix in a 9:1 molar ratio is ambiguous, given the manuscript does not mention the molecular weight of the alginate precursors.
(4) I found that certain key literature supports were missing in a few statements in the introduction section, including (a) “Stimuli-responsive hydrogels, such as glucose-responsive, pH-responsive, and thermoresponsive hydrogels, have been utilized to achieve controlled and sustained release of active pharmaceutical ingredients, proteins, and peptides.” This reference is only a pH-responsive example. (b) “One such polymer, SA, has a high capacity to chelate divalent cations like Ca2+, leading to the formation of a 3D network with a distinctive “egg-box” structure.” (c) “This pH-responsive behaviour is responsible for the effectiveness of alginate hydrogels as carriers for the oral administration of chemotherapeutics, water-insoluble drugs, and proteins/peptides.” (d) “Pluronic, a group of block copolymers with both hydrophilic and hydrophobic components, have gained widespread use in medical and pharmaceutical applications, due to their remarkable biocompatibility and low immunogenicity.” (d) The description of acetylsalicylic acid requires additional literature support.
(5) The reference section of the manuscript needs a thorough revision. The format is inconsistent, and certain references (such as ref 5 and ref 21) lack important details.
(6) While the manuscript is largely free of grammatical errors, the narrative flow requires substantial improvement. For instance, pages 7 to 8 have an abrupt insertion of sentences about F127 in the discussion on the swelling behaviors of the alginate beads, disrupting the flow of the text. A similar issue occurs on pages 9 to 10, where an extensive paragraph introduces acetylsalicylic acid.
Please see the comments above, thank you.
Author Response
Reviewer 3#
(1) The chemical composition of the alginate beads needs to be examined. The current term “sodium-alginate beads” is inappropriate, considering that the sodium alginate solution undergoes ion exchange with Ca2+ chelating the carboxylates of alginate when dropped into CaCl2.
According to the reviewer’s comment, in the 2.1 Materials, we added: “…..Alginic acid sodium salt from brown algae, composed of D-mannuronic and L-guluronic acid chains”
We also changed the term “sodium-alginate beads” with the correct one “alginate beads”
(2) The manuscript fails to mention the concentration of F127. The term “appropriate amounts of Pluronic...” is vague. The authors should provide quantitative details.
In the section 2.2.1 Preparation and physicochemical characterization of micelles, we added: “The colloidal concentration of Pluronic® F-127 is 100mg/ml, 10% w/v”.
(3) The claim that alginate and F127 mix in a 9:1 molar ratio is ambiguous, given the manuscript does not mention the molecular weight of the alginate precursors.
We made the correction: “9:1 weight ratio”
(4) I found that certain key literature supports were missing in a few statements in the introduction section, including (a) “Stimuli-responsive hydrogels, such as glucose-responsive, pH-responsive, and thermoresponsive hydrogels, have been utilized to achieve controlled and sustained release of active pharmaceutical ingredients, proteins, and peptides.” This reference is only a pH-responsive example. (b) “One such polymer, SA, has a high capacity to chelate divalent cations like Ca2+, leading to the formation of a 3D network with a distinctive “egg-box” structure.” (c) “This pH-responsive behaviour is responsible for the effectiveness of alginate hydrogels as carriers for the oral administration of chemotherapeutics, water-insoluble drugs, and proteins/peptides.” (d) “Pluronic, a group of block copolymers with both hydrophilic and hydrophobic components, have gained widespread use in medical and pharmaceutical applications, due to their remarkable biocompatibility and low immunogenicity.” (d) The description of acetylsalicylic acid requires additional literature support.
According to the reviewer’s comment, we added the appropriate literature in all the above sentences. For (a) we added the new reference:
Jiang Y, Wang Y, Li Q, Yu C, Chu W.Curr Med Chem. 2020;27(16):2631-2657. doi: 10.2174/0929867326666191122144916.Natural Polymer-based Stimuli-responsive Hydrogels.
(5) The reference section of the manuscript needs a thorough revision. The format is inconsistent, and certain references (such as ref 5 and ref 21) lack important details.
We corrected the format of all references.
(6) While the manuscript is largely free of grammatical errors, the narrative flow requires substantial improvement. For instance, pages 7 to 8 have an abrupt insertion of sentences about F127 in the discussion on the swelling behaviors of the alginate beads, disrupting the flow of the text. A similar issue occurs on pages 9 to 10, where an extensive paragraph introduces acetylsalicylic acid.
According to the reviewer’s comment, we made significant improvements in order to avoid misunderstandings.

Reviewer 4 Report
The research addresses an important issue related to the Design and development of low and medium-viscosity alginate 2 beads loaded with Pluronic® F-127 nano-micelles. The topic is part of the wider subject of the drug delivery challenges and is relevant for the Journal readers. The authors supported data for application, Before considering for publication, some issues need to be addressed.
1. the manuscript would benefit from an English correction.
2. the stability of nano nano-micelles is poor, as far as I know, the zeta potential for Stable particles must be more than 30mv, how can you confirm the stable formation of carriers?
3. why did the authors test the release profile in 180 min? and, is the nanocarriers long-term release?
4. As the nano-micelles are for treatment usage, the in vitro and in vivo tests should be added to better understand the cytotoxicity and efficiency of the nanocarriers.
5. It is better to add a graphical scheme for the preparation section.
6. more characterization such as FTIR,... are needed to better understand of nano-micelles' structure.
Moderate editing of the English language is required.
Author Response
- the manuscript would benefit from an English correction.
According to the reviewer’s comment, we made significant English language improvements.
the stability of nano nano-micelles is poor, as far as I know, the zeta potential for Stable particles must be more than 30mv, how can you confirm the stable formation of carriers?
According to the reviewer’s comment, we added justification about the stability of micelles:
“We also checked the colloidal stability of the Pluronic® F-127 nanomicelles with and without the encapsulated ASA for a period of 30 days. Both dispersions were stored at 4°C (in the fridge). The results from the physicochemical characterization after 30 days showed that both the nanomicellar formulations are stable (Dh≈140nm for pure micelles and Dh≈125nm for ASA micelles; the PDI values are slightly increased, and the ζ-potential remained unaffected). The last observation suggests that the steric stabilization due to the hydrophilic chains of the triblock copolymer is responsible for keeping the micelles in dispersion state, without the presence of electrostatic repulsion.”
why did the authors test the release profile in 180 min? and, is the nanocarriers long-term release?
According to the reviewer’s comment, in the section of release studies we added: “We performed the experiments for 180 min because during this time period all the encapsulated ASA was released. In our opinion, this release profile is a combination of the pH-responsive behavior of alginate beads and the long-term release of the Pluronic® F-127 nanomicelles. The nanomicelles act as a "second" barrier that should be destroyed for the release of the encapsulated ASA.”
As the nano-micelles are for treatment usage, the in vitro and in vivo tests should be added to better understand the cytotoxicity and efficiency of the nanocarriers.
According to the reviewer’s comment, we emphasized the safety profile of the used micelles in the introduction section. Namely, we added: “Pluronics, a group of block copolymers with both hydrophilic and hydrophobic components, have gained widespread use in medical and pharmaceutical applications, due to their remarkable biocompatibility, safety, and low immunogenicity. Pluronic F127, also named Poloxamer 407, is an FDA-approved temperature-sensitive hydrogel that has been widely used for drug delivery purposes and wound healing applications.”
It is better to add a graphical scheme for the preparation section.
According to the reviewer’s comment, we added a schematic representation of the formulations of our study.
Scheme 1. Schematic representation of A. Pluronic® F127 micelles, B. network of alginate beads, C. the formation of alginate beads, D. Pluronic® F127 micelles loaded with ASA and E. alginate beads encapsulated with Pluronic® F127 micelles loaded with ASA.
- more characterization such as FTIR,... are needed to better understand of nano-micelles' structure.
As mentioned in the introduction, Pluronics are widely used copolymers. The structure and morphology of nanomicelles are well established in the literature. In this section, several references are used to describe their physicochemical characteristics. In this study, we probed the physicochemical characteristics of the micelles, tested the influence of ASA on these characteristics, and evaluated the colloidal stability of all the micellar systems.

Round 2
Reviewer 1 Report
The authors have revised the manuscript substantially, therefore I would recommend it for publication.
Just a minor note: In a schematic diagram, the font size should be increased for better visualization.
Reviewer 4 Report
The paper can be accepted for publication without any further changes required from the authors.